# Apoptosis Genes as a Key to Identification of Inverse Comorbidity of Huntington’s Disease and Cancer

**DOI:** 10.3390/ijms24119385

**Published:** 2023-05-27

**Authors:** Elena Yu. Bragina, Densema E. Gomboeva, Olga V. Saik, Vladimir A. Ivanisenko, Maxim B. Freidin, Maria S. Nazarenko, Valery P. Puzyrev

**Affiliations:** 1Research Institute of Medical Genetics, Tomsk National Research Medical Centre, Russian Academy of Sciences, 634050 Tomsk, Russia; elena.bragina72@gmail.com (E.Y.B.); densema.gomboeva@medgenetics.ru (D.E.G.); maria-nazarenko@medgenetics.ru (M.S.N.); v.puzyrev@medgenetics.ru (V.P.P.); 2Institute of Cytology and Genetics, Siberian Branch of Russian Academy of Sciences, 630090 Novosibirsk, Russia; saik@bionet.ncs.ru (O.V.S.); salix@bionet.nsc.ru (V.A.I.); 3Department of Biology, School of Biological and Behavioural Sciences, Faculty of Science and Engineering, Queen Mary University of London, London E1 4NS, UK; 4Centre of Omics Technology, I.M. Sechenov First Moscow State Medical University, 119991 Moscow, Russia; 5Department of Medical Genetics, Faculty of General Medicine, Siberian State Medical University, 634050 Tomsk, Russia

**Keywords:** inverse comorbidity, dystropy, Huntington’s disease, neurodegenerative disease, apoptosis, cancer, gene network

## Abstract

Cancer and neurodegenerative disorders present overwhelming challenges for healthcare worldwide. Epidemiological studies showed a decrease in cancer rates in patients with neurodegenerative disorders, including the Huntington disease (HD). Apoptosis is one of the most important processes for both cancer and neurodegeneration. We suggest that genes closely connected with apoptosis and associated with HD may affect carcinogenesis. We applied reconstruction and analysis of gene networks associated with HD and apoptosis and identified potentially important genes for inverse comorbidity of cancer and HD. The top 10 high-priority candidate genes included *APOE*, *PSEN1*, *INS*, *IL6*, *SQSTM1*, *SP1*, *HTT*, *LEP*, *HSPA4*, and *BDNF.* Functional analysis of these genes was carried out using gene ontology and KEGG pathways. By exploring genome-wide association study results, we identified genes associated with neurodegenerative and oncological disorders, as well as their endophenotypes and risk factors. We used publicly available datasets of HD and breast and prostate cancers to analyze the expression of the identified genes. Functional modules of these genes were characterized according to disease-specific tissues. This integrative approach revealed that these genes predominantly exert similar functions in different tissues. Apoptosis along with lipid metabolism dysregulation and cell homeostasis maintenance in the response to environmental stimulus and drugs are likely key processes in inverse comorbidity of cancer in patients with HD. Overall, the identified genes represent the promising targets for studying molecular relations of cancer and HD.

## 1. Introduction

Cancer and neurodegenerative disorders are the most widespread and devasting pathologies in the world. Interestingly, epidemiological studies have shown a decrease of risk of cancer in patients with neurodegenerative diseases (NDDs), including Huntington disease (HD), Alzheimer’s disease (AD), and Parkinson’s disease [1,2]. The phenomenon of lowering the co-occurrence of two diseases in one person is known as inverse comorbidity [3] or dystropy [4].

The relationship between neurodegeneration and cancer remains an area under debate. The research of molecular mechanisms of inverse comorbidities is based on the search for common genes with opposite regulation [5]. Common key biological pathways that are important in the pathogenesis of both pathologies can impact the generation of inverse comorbidity. Processes such as cell cycle control, DNA reparation, immune signal transduction, and mitochondrial activity are among those that mainly affect the alternative behaviour of cells in relation to cell growth and survival under conditions of neurodegeneration and cancer [6].

Cell proliferation and apoptosis require adequate balance, since their dysregulation, caused by the dysfunction of relative mediators, can be considered a vital component of the inverse comorbidity of cancer and NDDs. Decreased activity and evasion of apoptosis are hallmarks of cancer cells [7]. On the contrary, enhanced apoptosis and loss of cells are characteristic for NDDs, including HD [8].

HD is an inherited neurodegenerative disorder caused by the expansion of CAG trinucleotide repeats in the first exon of the *HTT* gene (4p16.3); genes involved in DNA repair and mitochondrial dysfunction can modify disease pathogenesis [9]. HD is characterized by neurodegeneration, specifically of striatal spiny neurons. Clinically, HD involves a triad of symptoms: motor, psychiatric, and cognitive (non-motor). Epidemiological studies have shown a significantly lower cancer incidence among HD patients. Most have shown reduction in the overall cancer incidence [2,10,11,12,13,14]. The lowest cancer incidence was shown for breast cancer [2,10,12], lung cancer [2], colorectal cancer [2,10], malignancies of hematopoietic and lymphoid tissues [2,12], malignancies of the female reproductive system, and prostate cancer [10,12]. Skin cancer, on contrary, showed a higher incidence [11].

Assuming that *HTT* expansion is the main factor in HD, most studies on the inverse comorbidity of HD and cancer are focused on the *HTT* gene, the huntingtin protein, and CAG repeats. In mice models, mutant huntingtin was shown to accelerate breast cancer development and metastasis, reducing the endocytosis of the ErbB2/HER receptor kinase [15]. Wild-type huntingtin was shown to inhibit breast cancer metastasis by regulating expression and localization of tight junction protein ZO1. The same study showed that downregulation of *HTT* mRNA and protein can be a biomarker for breast cancer metastasis [16]. Assessment of the association of the *HTT* CAG repeat with cancer incidence revealed that the length of *HTT* CAG correlates with lower incidence of ovarian cancer in carriers of the *BRCA2* mutation and that CAG repeat length in the long *HTT* allele can be a factor in metastasis in sporadic breast cancer (HER+ subtype) [17]. One of the possible mechanisms linking HD and cancer may be RNA toxicity. Murmann A.E. and colleagues constructed small interfering RNAs based on *HTT* CAG repeats; these siRNAs exhibited toxicity toward cancer cells in vitro and in vivo [18].

The molecular mechanisms of the inverse comorbidity are still unknown. The majority of previous studies are more focused on the search for common and specific genes; however, this approach does not reveal even a small part of existing interconnections between the diseases. Such a reductionist approach is likely useful on the clinical level, but it ignores the fact that borders between disorders are not clear in terms of genetics and proteomics [19]. This impedes the search for possible pathways that cause or have an effect on the associations between diseases. One of the possible approaches is network analysis, which allows consideration of a wide range of different interactions between entities. Associative genetic networks allow for the study of complex diseases as well comorbidities [20]. A few bioinformatics and computational biology studies have been conducted to study the inverse comorbidity between cancer and NDDs [5,21,22,23].

Apoptosis plays an important physiological role in neurogenesis and carcinogenesis. The genes involved in apoptosis are considered as participants in the inverse comorbidity of cancer and NDDs [24]. Here, we explore apoptosis as one of the pathways linking HD and multiple cancer types for gene/protein network reconstruction to identify candidate genes underlying the inverse comorbidity of these diseases.

## 2. Results

### 2.1. Associative Networks and Gene Prioritization

Firstly, using ANDSystem, we reconstructed the associative network of HD. This associative network included 139 genes/proteins associated with HD (Appendix A). Interactions in the HD gene/protein network were represented by: association (244), involvement (13), regulation (11), and treatment (10). Interactions between the entities are represented by a wide range of processes, such as physical interaction between the proteins (e.g., cleavage of the protein by the protease, its degradation) or regulatory interaction, such as regulation of expression or regulation of the transport of proteins. Moreover, interactions in the ANDSystem also involve the presence of associations between entities (e.g., the association of the *HTT* gene with HD).

Then, the list of apoptosis-related genes was retrieved from the KEGG database (according the hsa04210 identifier). This list included 136 genes (Appendix A).

Ten genes, *AIFM1*, *CASP3*, *CASP6*, *CASP8*, *CASP9*, *CYCS*, *DIABLO*, *NFKB1*, *NGF*, and *TNF*, were shared between the associative networks of HD and the list of apoptosis-related genes. Inasmuch the aim was to evaluate the closeness of genes/proteins associated with HD according to their closeness to genes implicated in the process of apoptosis, these genes (apoptotic) were excluded from the associative network of HD.

Finally, we used 129 genes/proteins of the associative network of HD as the test set and 126 apoptosis-related genes as the training set in subsequent candidate gene prioritization. The results for genes according to the different prioritization criteria and the full list of the prioritization rank meanings of all HD network genes are listed in Appendix A. Among the top 10 highest priority genes are the following: *APOE*, *PSEN1*, *INS*, *IL6*, *SQSTM1*, *SP1*, *HTT*, *LEP*, *HSPA4*, and *BDNF*. These genes are presented in Table 1.

The most significant biological processes of the top 10 genes included learning and memory processes (GO:1902952, GO:1900273), amyloid plaque formation, and pathogenesis of particular central nervous system diseases (GO:1905908, GO:0042982, GO:1902004) (Appendix A).

Interestingly, the top 10 high-priority genes were tightly interconnected in the associative network of HD; moreover, these genes had many interactions with the apoptotic genes. Analysis of the final associative network determined that the majority of genes (*n* = 78) were implicated in apoptosis, which is reasonable because these genes were used for the gene network reconstruction. These top 10 high-priority candidate genes for inverse comorbidity and apoptotic genes (*n* = 78) comprised the final associative network, which consisted of 88 genes (Appendix A).

### 2.2. KEGG Functional Enrichment Analysis

KEGG enrichment analysis was carried out for the 88 genes of the final associative network (Appendix A). The most significant KEGG pathway was expectedly related with apoptosis (hsa04210; 78 genes; log(q-value) = −95.7). Fifteen KEGG pathways were linked with different cancer types. It is worth noting that among the significant pathways where the studied genes were overrepresented were such pathways as: lipid and atherosclerosis (hsa05417; 49 genes; log(q-value) = −78,8), the TNF signaling pathway (hsa04668; 32 genes; log(q-value) = −52.9), the neurotrophin signaling pathway (hsa04722; 30 genes; log(q-value) = −47.6), pathways of neurodegeneration—multiple diseases (hsa05022; 47 genes; log(q-value) = −40.0), and the MAPK signaling pathway (hsa04010; 32 genes; log(q-value) = −38.7) (Appendix A).

The final genes were involved in the following KEGG pathways: Huntington’s disease (hsa05016), pathways of neurodegeneration—multiple diseases (hsa05022), and cancer pathways (hsa05200). We found that 17 genes were enriched in the Huntington’s disease (hsa05016) KEGG pathway; among them, 11 genes (*TRAF2*, *MAPK8*, *BAX*, *CASP9*, *SP1*, *BBC3*, *CASP3*, *TP53*, *APAF1*, *CYCS*, and *CASP8*) were shared between Huntington’s disease (hsa05016) and pathways in cancer (hsa05200) (Figure 1). It total, 32 genes, which included *IL6* and *SP1* of the top 10 high-priority genes, were common for pathways in cancer (hsa05200) and pathways of neurodegeneration (hsa05022).

### 2.3. Genes of the Final Associative Network in GWAS

The results showed that six genes (*APOE*, *IL6*, *SQSTM1*, *CYCS*, *HTT*, and *CTSB*) of the final associative network (*n* = 88), according to the GWAS (DisGeNET database), are associated with NDDs (Appendix A). Fifteen genes are associated with cancer (*APOE*, *HSPA4*, *TP53*, *AKT2*, *AKT3*, *CASP8*, *AKT1*, *BCL2*, *FAS*, *BCL2L11*, *BAK1*, *ATM*, *PARP1*, *BID*, and *PIK3CD*).

Nine genes (*CASP8*, *ATM*, *TP53*, *HSPA4*, *BCL2L11*, *AKT1*, *BAK1*, *BCL2*, and *APOE*) were involved in various carcinomas at different localizations (Appendix A). At least three genes from this list were involved in the formation of carcinomas, including breast carcinoma (*CASP8*, *HSPA4*, *BCL2L11*), prostate carcinoma (*BCL2*, *CASP8*, *BCL2L11*, *ATM*), and chronic lymphocytic leukemia (*BCL2*, *CASP8*, *FAS*, *BCL2L11*, *BAK1*).

Three genes of the final associative network, *CASP8*, *ATM*, and *PARP1*, are associated in GWAS with melanoma as an example of positive associations of cancer with neurodegenerative disorders characterizing different molecular pathophysiology (Appendix A).

A substantial number of genes (*n* = 35: *APOE*, *IL6*, *CTSB*, *HSPA4*, *TP53*, *AKT2*, *AKT3*, *CASP3*, *CASP8*, *CASP9*, *BCL2*, *FAS*, *ATM*, *PARP1*, *BID*, *IKBKB*, *TRAF1*, *TRAF2*, *PIK3CA*, *APAF1*, *MCL1*, *NFKBIA*, *NGF*, *RAF1*, *CHUK*, *RIPK1*, *MAP3K5*, *ENDOG*, *BDNF*, *PIK3CB*, *DFFA*, *SP1*, *MAPK1*, *NRAS*, *NFKB1*) were associated with endophenotypes (systolic/diastolic pressure, body mass index, C-reactive protein, serum cholesterol level, smoking behavior), which can be considered as risk factors of neurodegenerative diseases/cancer. 

### 2.4. Expression of Genes of the Final Associative Network in the Blood of Patients with HD and Breast/Prostate Cancer

Identification of differentially expressed genes among genes of the final associative network was conducted by RNA-seq analysis, single cell RNA sequencing, and high throughput sequencing in peripheral blood mononuclear cell patients with HD and breast and prostate cancer. Only one gene, *AKT3*, was down-regulated in the blood of HD patients compared with healthy individuals (logFC = −0.39; *p* value = 0.0374). However, this difference was not present after multiple comparison corrections (Appendix A).

In addition, it is worth noting some minor differences in expression of genes between datasets. The expression of three genes (*BCL2L11*, *PIK3CA*, and *XIAP*) regulated in opposite directions (up/down) occurred in patients with HD and breast/prostate cancer (Appendix A). The expression of 24 genes was regulated in opposite directions in patients with HD and breast cancer (Appendix A). Seven genes (*ATM*, *BCL2*, *ERN1*, *FOS*, *NRAS*, *PIK3R1*, and *SP1*) were regulated in opposite directions in patients with HD and prostate cancer. However, these differences were not statistically significant.

### 2.5. Identifying Tissue-Specific Functional Modules of Genes of the Final Associative Network

Genes of the associative network are multifunctional, and their effects depend on functional partners in various tissues. Thus, we considered the specificities of gene interactions in tissues, which are predominantly affected in HD and cancers (Appendix A). We selected the mammary epithelium and prostate gland for the analysis, because the respective types of cancer (breast and prostate cancer) are in inverse comorbidity with HD [10,12]. HD predominantly affects the basal ganglia; thus, we compared the obtained results of functional modules for selected tissues with the basal ganglia.

The HumanBase functional gene module analysis was used for identifying cohesive gene clusters and for representing the local gene network neighborhood for the genes of the final associative network [25]. Thus, 76 of the 88 genes of the final associative network were assigned to 9 modules in the basal ganglion, 74 genes to 10 modules in the mammary epithelium, and 78 genes to 9 modules in the prostate gland.

Generally, as expected, the products of genes of the final associative network in all considered tissues are involved in apoptosis signaling regulation. They are also important for other vital cellular functions, such as inflammatory response, mitochondrion organization, collagen synthesis, regulation of DNA binding transcription factor activity, miRNA metabolic processes, and others (Appendix A).

There are some specific differences in the gene interactions depending on the tissue context. *APOE* and *NFKB1* in the basal ganglia cooperate to perform functions associated with the metabolism of lipids and the reaction of hormones in response to toxic exposure (Appendix A). In mammary tissue, unlike basal ganglia, these functions are realized by only *INS* and *TNF*.

*HTT* is involved in the modules with similar functions but interacts with slightly different entities. For the cellular response to the various stimuli, including to oxidative stress and starvation, *HTT* interacts directly with *MAPK3*, and through an intermediary, with *MAP3K5* and *PARP1* in the basal ganglia (Appendix A). However, *MAP3K5* and *TNFRSF1A* are partners of *HTT* in the mammary epithelium. Four proteins (*ATM*, *PIK3R2*, *PSEN1*, and *PIK3CD*) interact with *HTT* in the prostate gland.

In all analysed tissues, genes of mitogen-activated protein kinases were presented in the functional module to link with the cell response to external stimuli, including drugs. In the basal ganglia, this function is exerted by *ATF4*, *MAPK3*, *NFKB1*, *RIPK1* and *BAK1*; in the prostate gland—*CHUK*, *MAPK1*, *MAPK3*, *MAPK8*, *ATF4*, *BAK1*, *NFKB1*; in the mammary epithelium—*MAP3K5*, *TNFRSF1A*. Involvement of mitogen-activated protein kinases in the cascades of cell reactions, caused by external stimuli and participation in cell fate-determining signalling pathways (such as differentiation and cell survival), makes these genes potentially meaningful in the inverse comorbidity of cancer and neurodegeneration.

## 3. Discussion

The rare co-occurrence of diseases represents a scientific enigma. Inverse comorbidity is caused by biological phenomena [3]. While the growing number of epidemiological studies has pointed to the existence of inverse comorbidity of neurodegenerative diseases and cancer [1,2], there remains a lack of knowledge on which genetic factors clearly contribute to this phenomenon.

Taking into the account the heterogeneity and the redundancy of the factors that play a role in the pathogenesis of cancer and the excessive knowledge on the subject, we reduced the number of model processes. Relying on the hypothesis regarding the role of apoptosis in the regulation of HD and cancer, we used the bioinformatics approach for associative network reconstruction. Using ANDSystem [26,27], we generated the associative network of HD and then retrieved the apoptosis-related genes from KEGG. Then, we performed gene prioritization using ToppGene and special criteria of ANDSystem. Finally, we combined the associative network of HD with apoptosis genes and identified the top 10 highest priority genes potentially relevant to inverse comorbidity or dystropy of cancer and HD.

The top 10 genes (*APOE*, *PSEN1*, *INS*, *IL6*, *SQSTM1*, *SP1*, *HTT*, *LEP*, *HSPA4*, and *BDNF)* comprise high-priority candidate genes for inverse comorbidity. The role in inverse comorbidity of cancer and Parkinson’s disease/Alzheimer’s disease was shown for *APOE*, *APOC1*, *SQSTM1*, *PSEN1*, and *SP1* [22,28,29,30]. However, this is the first study to report their likely role in the dystropy of cancer and HD.

Some of the top 10 genes (*LEP*, *IL6*, *APOE*, *INS*) are involved in metabolic pathways of lipids (Appendix A). Peripheral metabolic dysfunction related to weight loss in patients with HD is considered as a direct effect of mutant *HTT*, which is expressed in many tissues, including fat tissue [31]. This was confirmed by experiments that showed high levels of lipids in the body fat of HD *Drosophila* and large number of depolarized mitochondria [32,33] at both early and late stages of the disease. The excess of lipids and damaged mitochondria activate autophagy through the regulatory system. Meanwhile, activation of cytotoxic autophagy is considered unfavorable for cancer cells due to the induction of cell death [34].

It worth noting that peripheral metabolic dysfunction is a feature not only of HD, but also of other disorders [35] that inversely correlate with cancer, namely AD and PD. This fact points to the presence of multiple effects of other proteins, similar to *HTT* or proteins interacting with them. This circumstance does not exclude the participation of other regulatory systems [36], because both wild-type and mutant *HTT* forms interact with many proteins in the human interactome (2631 and 2740 respectively) [37].

The rest of the top 10 high-priority genes have not been fully investigated in the aspect of inverse cancer comorbidity in patients with neurodegenerative diseases. However, they might have important roles in the inverse association of cancer and HD, according to biological aspects.

Protein homeostasis plays a pivotal role in the pathophysiology of cancer and neurodegenerative diseases. Such genes as *HSPA4* encode heat shock protein family A member 4 of the Hsp70/Hsp100 family, which functions as a chaperon and ATP-binding nucleotide exchanging factor [38]. Interacting with transcription factor ZONAB, *HSPA4* is able to control cell proliferation [39]. The role of *HSPA4* was shown both for cancer and neurodegenerative diseases. *HSPA4* is overexpressed in cancers [40]. The complex of Hsp70, Hsp110 (*HSPA4*), and J-protein suppresses the amyloid formation of HttQ_n_ fibrils in HD patient-derived neural progenitor cells and on an organismal level in *Caenorhabditis elegans* [41].

At the level of neurons, *BDNF*, which encodes brain derived neurotrophic factor [42], contributes to the metabolism and trophic functions [43]. Wild-type huntingtin regulates *BDNF* transcription [44] and its vesicular transport from the cortex to the striatum [45]. Mutant huntingtin leads to reduced *BDNF* transcription [44], impaired vesicular transport [46], and inhibited secretion [47]. Due to the trophic functions, *BDNF* is also implicated in tumor development, progression, and survival [48]. *BDNF* expression is up-regulated in epithelial cells of mammary gland tumors [49] and biopsy samples of prostate glands [50].

The top10 high-priority candidate genes for inverse comorbidity were highly interconnected in the associative network of HD and interacted with the apoptotic genes. We analyzed the final associative network that includes these interacting genes. They were highly enriched with processes linked with both cancer and neurodegeneration (Appendix A).

It is important to note that the majority of studies concerning inverse comorbidity of cancer and neurodegenerative diseases are focused on the research of regulation changes of main signal pathways of cancer and neurodegeneration, mostly through the gene interactions in gene networks and through the dysregulation of basic cell processes [5,21,22,23].

Murmann A.E. et al. [18] demonstrated the toxic effect of small interfering RNAs, based on trinucleotide CAG-repeats, on cancer cells. These results reflect a substantial biological role of pathogenic expansion of trinucleotide CAG-repeats in inverse cancer comorbidity.

The role of specific involvement of each of the genes that are affected by pathogenic expansion is a current topic of research. Unstable trinucleotide repeat expansions affect many of disease-associated genes, though the link between neurodegenerative diseases is still poorly studied at present [18]. Thus, the role of these disease-associated genes regarding inverse association with cancer is a subject of future research.

According to GWAS, *HTT* gene haplotypes, which span the pathogenic expansion region, are associated only with a single feature—HD age of onset [51]. However, no one GWAS found the association of polymorphisms of the *HTT* gene with any cancer.

Despite the fact that only a limited number of genes are associated with a phenotype (AD, PD, different cancer types), many genes of the final associative network are associated with risk factors and endophenotypes of these disorders, including smoking, body mass index, serum LDL cholesterol measurement, and systolic/diastolic pressure (Appendix A).

More than one gene of the final associative network is significantly differentially expressed (with multiple comparison correction) in the peripheral blood of HD/breast/prostate cancer patients compared with healthy individuals. In this area of study, we face the serious problem of the absence of relevant data sets for inverse comorbidity of HD and cancer. Thus, the gene expression study of candidate genes of inverse comorbidity in human target organs affected in cancers is almost impossible due to low prevalence of HD, very low cancer incidence in HD patients, and limited availability of biological material during the patient lifetime. The only available material that can be used for our research aims is whole blood, as blood links all relevant tissues. Though it is unclear how blood reflects the changes in HD, this question is still important and should be considered in further studies. However, the further analysis of tissue-specific functionality of the studied genes of the final associative network revealed their involvement in similar functional modules, with some exceptions. Several functional modules, which consist of genes of the final associative network, are linked with cell response to the external stimuli; it is likely that drugs and other external factors influence the decrease of risk of carcinogenesis in neurodegenerative diseases. The role of drugs in the activation of mechanisms of tumor growth, in particular melanoma, is discussed regarding levodopa treatment in PD [52].

## 4. Materials and Methods

In this study, we used the hypothesis of the important role of apoptosis as key process in neurodegeneration and carcinogenesis; regulation of apoptosis can be one of the possible mechanisms of inverse comorbidity of cancer and HD. Consequently, genes/proteins that are more tightly linked with the apoptosis present a significant research area as they can be involved in the regulation of tumour-related processes.

The study was implemented in several phases. The flowchart of the study is presented in the Figure 2.

### 4.1. Reconstruction of Genetic Associative Networks

The reconstruction of associative networks was performed using text-mining ANDSystem software (http://www-bionet.sscc.ru/and/cell/) [26,27].

ANDSystem is a tool for the reconstruction of associative networks, which comprise two modules: ANDCell for the knowledge extraction (located on the server) and ANDVisio for the visualization (the client module) [26]. ANDCell extracts knowledge from the scientific literature (PubMed) and biological and medical databases (SwissProt, Entrez GENE, ChEBI, MESH, MirBase, Gene Ontology, Cell Lines Database (CLDB), Entrez Taxonomy; IntAct, MINT, TRRD, EntrezGene, UniProtGOA) using the text-mining method. ANDVisio automatically reconstructs and visualizes networks [26]. ANDSystem allows retrieval of the description of the wide scope of interaction between such entities as genes, proteins, diseases, pathways, and cellular components. ANDSystem has been used for retrieving candidate genes of asthma and tuberculosis [53] and for the analysis of neuronal apoptosis genes in the associative network of Parkinson disease [54].

In the current study we reconstructed two associative gene networks using ANDSystem software: (1) the associative network for HD; (2) the final associative network. In order to reconstruct the associative network of HD, the keyword “Huntington disease” was used.

The reconstructed associative network of HD included all genes and proteins associated with HD (*n* = 139; Figure 2). The final associative network included the top 10 genes from the associative network of HD and the 78 apoptosis-related genes interacting with them, as retrieved from the KEGG database.

### 4.2. Prioritization of Candidate Genes of Inverse Comorbidity

The set of genes implicated in apoptosis was retrieved from the KEGG database by the hsa04210 identifier [55]. All genes and proteins of the associated network of HD were considered as the test set of candidate gene prioritization for inverse comorbidity, except 10 genes, which were identified among the set of apoptosis-related genes. The apoptotic gene list was considered as the training set in gene prioritization (*n* = 136; Figure 2). Then, the gene prioritization was performed using ANDSystem and ToppGene [56]. For each of the candidate genes, the following six criteria were calculated: betweenness centrality, closeness centrality, stress centrality, cross-talk specificity (CTS), cross-talk centrality (CTC) using the ANDSystem tool, and priority parameters (using the ToppGene tool (https://ToppGene.cchmc.org). ToppGene tool prioritization criteria were calculated using HD genes/proteins as the test set and the apoptosis gene list as the training gene set. A more precise explain of these criteria can be found in other papers [54,57]. For a ranking of candidate genes from a test set according to the degree of their closeness to apoptosis-related genes (training gene set), mean rank was used, calculated according to the six abovementioned priority parameters [57].

### 4.3. Functional Enrichment Analysis

Analysis of KEGG pathway enrichment for the genes of the final associative network, which included the top 10 high-priority genes’ interaction with apoptotic genes (*n* = 88) was performed used the web-based Metascape tool [58] with default parameters. The KEGG enrichment analysis was performed to identify important biological processes related to genes of the final associative network (accessed on February 2023). The results are presented in Appendix A. *p*-values and False Discovery Rates (FDR) or BH-adjusted *p*-values (q-value) indicate the significant pathways. In Mediascape, *p*-values and q-values are provided on the log-10 scale (LogP and Log(q-value) respectively). Therefore, a more negative *p*-value indicates a lower probability for the observed enrichment to occur by chance.

### 4.4. Analysis of GWAS Results

We screened the final associative network genes (*n* = 88) in GWAS (Score_gda ≥ 0.1) through the DisGeNET database (http://www.disgenet.org) (accessed on 3 February 2023). We used DisGeNET because the database integrates data of genes and variants associated to human diseases from expert curated repositories and GWAS catalogues [59]. A query was made for the genes of the final associative network, which found the top 10 high-priority genes interacting with apoptotic genes (*n* = 88). The result is presented in Appendix A.

### 4.5. Identification of Differentially Expressed Genes of the Final Associative Network

In order to evaluate the expression of genes of the final associative network in blood of the patients with HD and cancer, we used the web tool GREIN, which allowed us to analyze GEO RNA-seq datasets [60]. We used expression datasets of the blood of HD patients and two cancer types, breast and prostate cancer, as these show lower incidence rates in HD patients [10,12]. Other criteria for the datasets included the presence of a control group and human origin of studied samples. Taking into account all the abovementioned criteria, we chose three data sets, the summary of which are presented in Table 2.

We analyzed expression genes of the final associative network from HD, breast cancer, and prostate cancer based on a threshold *p*-value and log fold change value. Upregulated were genes with *p*-value ≤ 0.05 and log fold change value < 1; downregulated were genes with *p*-value ≤ 0.05 and log fold change value < 1. The results are presented in Appendix A.

### 4.6. Functional Modules of the Final List of Genes

Functionally clustered modules in the final associative network in the mammary epithelium, prostate gland and basal ganglia were identified by the HumanBase public database (https://hb.flatironinstitute.org, accessed on 10 January 2023). HumanBase applies community detection to find cohesive gene clusters from a provided gene list and selected relevant tissue [25].

The approach is based on shared k-nearest-neighbors (SKNN) and the Louvain community-finding algorithm to cluster the user-selected tissue network into distinct modules of tightly connected genes. The SKNN-based strategy has the advantages of alleviating the effect of high-degree genes and accentuating local network structure by connecting genes that are likely to be functionally clustered together.

Significantly enriched Gene Ontology Biological Process (GOBP) terms of each module are presented. Significance was calculated using Fisher’s exact tests followed by Benjamini–Hochberg corrections. The results are presented in the Appendix A.

## 5. Conclusions

The main aim of this paper was to discover new targets for the research of the inverse comorbidity of HD and cancer. Here, using the associative gene network reconstruction and focusing on apoptosis as a key process of both cancer and neurodegeneration, bioinformatic tools were used to interpret the associations of identified genes related to the reverse comorbidity of cancer and HD [61]. We considered many genes, but using a specific approach of prioritization, we delimited the scope of the studied genes to the top 10 candidate genes (*APOE*, *PSEN1*, *INS*, *IL6*, *SQSTM1*, *SP1*, *HTT*, *LEP*, *HSPA4*, *BDNF*), which comprise candidate genes of inverse comorbidity of HD and cancer.

Apart from apoptosis, other processes, such as peripheral metabolic dysfunction, which are related to the top 10 high-priority candidate gene functions, also are important in the inverse comorbidity of cancer and neurodegenerative diseases. Drug response and response to other stimuli also can affect the risk of cancer during the treatment of patients with neurodegeneration. Therefore, the genes identified in this study as candidate genes are important for further exploration of the inverse comorbidity of cancer and HD. In this study, we retrieved only those genes and proteins associated with the apoptosis process; broader aspects can be considered in future. Further, it is plausible to consider other biological overlaps between cancer and neurodegeneration (other biological processes such as genes of invasion, metastasis, tumor suppressors, anti-oncogenes, genes of protein quality control, autophagy, etc.). This is the first study that identifies candidate genes of inverse comorbidity of cancer and HD other than *HTT*.

Experimental research of the phenomenon of inverse comorbidity of HD and cancer contributes not only to the discovery of the mechanisms underlying the development of these diseases, but also affects the treatment of diseases that are negatively associated at the phenotypic level. In recent years, many studies have been published on the potential for cancer drugs repurposed to treat diseases of the central nervous system [62]. For example, the antiparkinsonian drug selegiline has been found to cause transcriptomic changes opposite to those seen in some types of cancer [63]. Selegiline belongs to the class of drugs of monoamine oxidase inhibitors. Its function is preventing monoamine neurotransmitters from breaking down, which are responsible for inhibiting the histone lysine specific demethylase 1, which leads to cancer [64]. Therefore, it is expected to perform efficiently against different types of cancers.

## Figures and Tables

**Figure 1 ijms-24-09385-f001:**
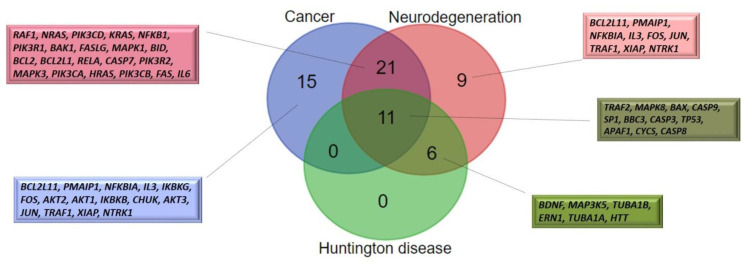
Comparison of the lists of genes enriched in KEGG pathways: Huntington’s disease (hsa05016), pathways of neurodegeneration—multiple diseases (hsa05022), pathways in cancer (hsa05200). Note: Huntington disease = Huntington’s disease (hsa05016); Cancer = pathways in cancer (hsa05200); Neurodegeneration = pathways of neurodegeneration—multiple diseases (hsa05022).

**Figure 2 ijms-24-09385-f002:**
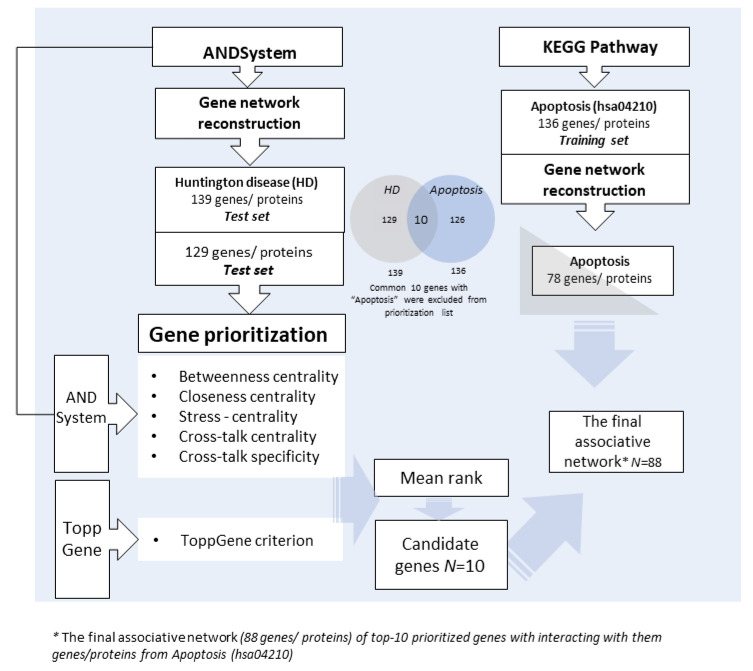
The flowchart of the study.

**Table 1 ijms-24-09385-t001:** Top 10 candidate genes/proteins, from a number of genes associated with Huntington’s disease, that have the highest priority according to closeness to genes of apoptosis.

Gene *	Chromosome	Entrez Gene ID	Protein Name
*APOE*	19	348	apolipoprotein E
*PSEN1*	14	5663	presenilin 1
*INS*	11	3630	insulin
*IL6*	7	3569	interleukin 6
*SQSTM1*	5	8878	sequestrum 1
*SP1*	12	6667	Sp1 transcription factor
*HTT*	4	3064	huntingtin
*LEP*	7	3952	leptin
*HSPA4*	5	3308	heat shock protein family A (Hsp70) member 4
*BDNF*	11	627	brain derived neurotrophic factor

Note: *—genes ranked according to the mean rank of prioritization criteria.

**Table 2 ijms-24-09385-t002:** Characteristics of gene expression data sets.

Study ID	Overall design	*N* (Case vs. Control)	Platforms
GSE61405	RNA-seq profiles of blood taken from normal controls and Huntington’s disease patients	11 vs. 8	GPL9115 Illumina Genome Analyzer II
GSE174431	RNA-Seq and single cell RNA sequencing of PBMCs from metastatic breast cancer patients	6 vs. 2	GPL24014 Ion Torrent S5 XL
GSE97901	Whole blood miRNA samples from both controls and patients where sequences and a differential expressional analysis was conducted to identify possible biomarkers to distinguish patients from controls	28 vs. 12	GPL11154 Illumina HiSeq 2000

## Data Availability

Not applicable.

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
