# Peer review of "Apoptosis Genes as a Key to Identification of Inverse Comorbidity of Huntington’s Disease and Cancer"

_ijms, 2023, doi:10.3390/ijms24119385_

Round 1

Reviewer 1 Report

While an increasing number of epidemiological studies have shown that there is an inverse comorbidity between neurodegenerative diseases (NDD) and cancer, there is still a dearth of understanding regarding which genetic factors clearly determine this phenomenon.

The present manuscript is a bioinformatics study uncovering particular genetic mechanisms, attributed to genes from the apoptotic pathway, underlying the inverse comorbidity of two principally different diseases, such as Huntington's disease and cancer. The authors proposed a hypothesis about the role of apoptosis as a key biological process that may contribute to neurodegeneration and carcinogenesis. The top-10 genes such as APOE, PSEN1, INS, IL6, SQSTM1, SP1, HTT, LEP, HSPA4, and BDNF were found to be high-priority candidate genes for inverse comorbidity (also known as dystrophy) between NDD and cancer. It is an interesting and well-conducted study, providing new insights into the molecular mechanisms underlying these disorders.  However, I have some comments that the authors should address when revising the manuscript.

1. In the KEGG Functional Enrichment Analysis, the authors provided log(q-values) to demonstrate the enrichment of selected genes with KEGG terms. It is unclear which log (q-values) may be considered statistically significant. I recommend providing uncorrected FDR values for this purpose.

2. At the end of the manuscript, I recommend that the authors write about the fundamental and applied significance of their research findings, as well as provide perspectives for future research.

3. The gene symbols should be italicized throughout the text.

Misc.:

The text of manuscript should be checked for grammatical and spelling errors as well as for typos, such as, for example, “Gee Ontology” (page 9, line 374).

 Citation “Greene C.S. et al., 2015” should be converted into a numeric 28 of the reference list.

The above comments should not be considered critical, and I believe that this manuscript is suitable for publication in IJMS, in the section "Molecular Genetics and Genomics", particularly in the special issue "Genes and Human Diseases".

The text of the manuscript should be checked for grammatical and spelling errors in English as well as for typos.

Author Response

  1. In the KEGG Functional Enrichment Analysis, the authors provided log(q-values) to demonstrate the enrichment of selected genes with KEGG terms. It is unclear which log (q-values) may be considered statistically significant. I recommend providing uncorrected FDR values for this purpose.

Remark taken into account. q values are FDR False Discovery Rate (FDR) or BH-adjusted p-values. The corresponding explanations are made in the text (section Materials and Methods):

P-values and False Discovery Rates (FDR) or BH-adjusted p-value (q-value) indicate the significant pathways. In Metascape p-values and q-values are provided on the log-10 scale (LogP and Log(q-value) respectively). Therefore, a more negative p-value indicates the less chance for the observed enrichment to occur by chance.

  1. At the end of the manuscript, I recommend that the authors write about the fundamental and applied significance of their research findings, as well as provide perspectives for future research.

Done. The following text has been added (section Conclusions):

Experimental research of the phenomenon of inverse comorbidity of HD and cancer contributes not only to the discovery of the mechanisms underlying the development of these diseases, but also affect the treatment of diseases that are negatively associated at the phenotypic level. In recent years, a lot of studies has been published on the potential for cancer drug repurposing used to treat diseases of the central nervous system [62]. For example, the antiparkinsonian drug selegiline has been found to cause transcriptomic changes opposite to those seen in some types of cancer [63]. Selegiline belongs to the class of drugs of monoamine oxidase inhibitors. Its function is preventing monoamine neurotransmitters from breaking down which are responsible for inhibiting the histone lysine specific demethylase 1, which leads to cancer [64]. Therefore, it is expected to perform efficiently against different types of cancers.

  1. The gene symbols should be italicized throughout the text.

Done.

Misc.:

The text of manuscript should be checked for grammatical and spelling errors as well as for typos, such as, for example, “Gee Ontology” (page 9, line 374).

Done.

Citation “Greene C.S. et al., 2015” should be converted into a numeric 28 of the reference list.

Done.

Reviewer 2 Report

This is an interesting study trying to search the molecular basis of inverse comorbidity in Huntington’s disease with several types of cancers.

The major concern of the study is that HD is a rare monogenic disease and cancer is not a single disease, and there are hundred of cancer types. The authors should modify the hypothesis and aim of the study. On lines 90-93, they state “The genes involved in apoptosis are considered as participants in the inverse comorbidity 91 between cancer and neurodegeneration”; however, HT and neurodegeneration are different concepts, and they can only state that the network analysis can be only interpreted in relationship with HD, not with neurodegeneration.

In addition, the inverse comorbidity between HD and several cancers is unidirectional since the difference in the prevalence of cancer and HD is so large that no cancer is associated with HD.

With this approach, the authors should revise the text and delete other references to other neurodegenerative disorders such as Alzheimer disease or Parkinson disease that have different mechanisms. This includes part of the supplementary tables such as Table S8.

Author Response

Thanks for this important note. Understanding the significant differences in the pathogenesis of both oncological disorders and diseases associated with neurodegeneration, we have tried to rewrite the text. We focused the text only on Huntington's disease and delete the text to the neurodegenerative disorders such as Alzheimer disease or Parkinson disease. Supplementary table 8 (The associations between SNPs rs405509, rs7412, and Alzheimer disease) was also excluded.

Round 2

Reviewer 2 Report

The authors have modified the manuscript limiting the inverse comorbidity approach to Huntingtong disease.